# DNN Model Compression Under Accuracy Constraints

## Abstract

The growing interest to implement Deep Neural Networks (DNNs) on resource-bound hardware has motivated the innovation of compression algorithms. Using these algorithms, DNN model sizes can be substantially reduced, with little-to-no accuracy degradation. This is achieved by either eliminating components from the model, or penalizing complexity during training. While both approaches demonstrate considerable compressions, the former often ignores the loss function during compression while the latter produces unpredictable compressions. In this paper, we propose a technique that directly minimizes both the model complexity and the changes in the loss function. In this technique, we formulate compression as a constrained optimization problem, and then present a solution for it. We will show that using this technique, we can achieve competitive results.

## 1 Introduction

Deep Neural Networks have been rapidly improving in many classification tasks, even surpassing human accuracy for some problems (Russakovsky et al., 2015). This high accuracy, however, is obtained by employing wider Zagoruyko & Komodakis (2016) or deeper He et al. (2016) networks. Some prominent networks today even surpass 100 layers, store hundreds of millions of parameters, and require billions of operations per input sample He et al. (2016); Srivastava et al. (2015). Such large networks are not well-suited for resource-bound, embedded and mobile platforms which are dominating the consumer market. This mismatch between computational requirements and available resources has motivated efforts to compress DNN models.

Compression techniques exploit the redundancy inherent to neural networks that emerges due to the considerable number of parameters in them. These many parameters help learn highly informative features during training. However, they simultaneously learn multitudes of unnecessary, ineffectual ones. Han et al. (2015) reduces these redundancies by pruning the network and quantizing the remaining parameters. Using these techniques, they were able to reduce the model size by more than an order of magnitude. Their success inspired other methodical approaches such as the soft weight-sharing (Ullrich et al., 2017). This method encodes model parameters using Bayesian prior and then penalizes this prior during training. As a result, it performs both pruning and quantization simultaneously and achieves superior compressions with negligible loss in accuracy.

In this work, we take a similar, integrated approach with the twist that we directly minimize the complexity. Unlike soft weight-sharing, however, we encode the parameters using the k-means objective which imposes less computations. We further apply a hard constraint on the training loss to maintain sufficient accuracy. Such a constraint takes advantage of the fact that we have already obtained some information about the loss function during training. We then present a straightforward solution for this constrained optimization problem. Our solution iteratively minimizes the k-means objective, while satisfying the loss constraint. Consequently, it can compress the model progressively where the compression rate can be adjusted. Finally, we test the proposed technique on three popular datasets and show that this method can achieve state-of-the-art compression with minimal loss of accuracy.

## 2 Related Work

Robustness of DNNs to noisy parameters is a byproduct of their redundancy and has been utilized frequently to facilitate and accelerate their implementations. Limited precision representations which introduce quantization noise, for example, have been widely used to simplify computations (Gupta et al., 2015) and design neural network hardware (Jouppi et al., 2017). Other works, take a different approach and eliminate most parameters (pruning) (Molchanov et al., 2016), aiming to reduce the total number of computations instead. Finally, weight-sharing techniques (Han et al., 2015) encode the remaining parameters of the networks in a small dictionary, reducing the storage requirement of the network. In this paper, we will be focusing on the last two techniques (pruning and weight-sharing), which were proposed first by Han et al. (2015). Our goal is to present a simple, unified method for compressing a trained DNN model.

Han et al. (2015) first proposed Deep Compression which introduced pruning and weight-sharing as techniques to compress DNN models. Pruning eliminates unnecessary network connections identified by their magnitude. Weight-sharing then encodes parameters into a small dictionary, using the k-means algorithm. This technique also employs an intermediate tuning step which offsets the effect of pruning on accuracy. However, the changes in the training loss function are, for the most part, ignored in this technique. Later studies demonstrated that taking this effect into account, either in pruning or quantization, can improve the compression. Guo et al. (2016), for example, formulated pruning as an optimization problem. Their method essentially performs pruning and tuning simultaneously, but does not perform weight-sharing. Likewise, Choi et al. (2016) incorporate this effect by augmenting the k-means objective by the training loss Hessian. However, they only consider the Hessian at one point, and their method requires prediction of the compression ratio beforehand.

Soft weight-sharing, proposed by Ullrich et al. (2017), minimized both training loss and model size simultaneously. Unlike the previous attempts, they encoded the network parameters using a Bayesian prior. This prior is then penalized when optimizing the training loss function and the mixing ratio controls the trade-off between accuracy and model size. This is then optimized using a SGD method. The limitation of this technique is the complexity of the Bayesian prior. Due to this limitation, in this paper, we will focus on k-means encoding of the weights, even though our method can be easily extended to Bayesian encoding as well. p

## 3 Method

The proposed algorithm compresses a trained neural network model by minimizing its complexity while constraining its loss function. Here, we first review the optimization process for the loss function during the training process. Then, we discuss the k-means encoding, which we use to measure the model complexity. Finally, we will present our compression algorithm.

### 3.1 Training Neural Networks

Training neural networks involves minimizing a loss function that quantifies the dissimilarity between the model output distribution and the ideal distribution. This optimization problem is formally written as:

$$\widehat{W} = \arg \min_{W} \mathcal{L}(W, \mathcal{X}, y) \tag{1}$$

Here, $\mathcal{X}$ and $y$ define a set of samples and ideal output distributions we use for training. $W$ is the learnt model and is the concatenation of all model parameters, ($W = [w_1 \ w_2 \ \ldots \ w_N]^T \in \mathbb{R}^N$ where $N$ is the number of parameters). $\widehat{W}$ is the final, trained model. Also, $\mathcal{L}(W, \mathcal{X}, y)$ represents the average loss over the dataset ($\{\mathcal{X}, y\}$) corresponding to the model $W$. This function is typically chosen as Cross entropy or hinge square loss. In the rest of this paper, we represent $\mathcal{L}(W, \mathcal{X}, y)$ as $\mathcal{L}(W)$ for simplicity. The local optimum for $W$ in this problem is usually found iteratively, using a SGD method.

## 3.2 K-MEANS ENCODING

The weight-sharing techniques proposed by Han et al. (2015) and Guo et al. (2016) use the k-means algorithm to encode the network parameters. Each parameter is then replaced by the nearest of the $k$ centroids. This procedure can also be defined as the following optimization problem:

$$\widehat{C} = \arg\min_{C} \sum_{i=1}^{N} \|w_i - m_i\|^2 \tag{2}$$

$$\forall i : m_i = \arg\min_{m \in C} \|w_i - m\| \tag{3}$$

where $M = [m_1 \ m_2 \ \ldots \ m_N]^T \in \mathbb{R}^N$ is the concatenated array of all centroids $m_i$ corresponding to the parameters $w_i$. The set $C = \{c_1, \ c_2, \ldots, \ c_k\} \subset \mathbb{R}$ also defines the set of cluster centroids. The commonly used heuristic solution to this problem is to iteratively find $M$ and update centroids accordingly. In each iteration $t$ this update is performed by:

$$c_j^{t+1} = \frac{1}{|J|} \sum_{i \in J} w_i \tag{4}$$

$$J = \{i | m_i = c_j^t\} \tag{5}$$

## 3.3 PROPOSED COMPRESSION METHOD

Similar to weight-sharing in Deep Compression (Han et al., 2015), we minimize the k-means objective function to reduce the model complexity. Further, similar to soft weigh-sharing (Ullrich et al., 2017), we minimize the changes in the loss function. However, we control these changes through a hard constraint to take advantage of the information already learnt through the training phase. In the rest of this section, we will formulate the problem of minimizing the complexity of the model and propose a solution for it.

As previously discussed, we use the k-means objective to represent the complexity of the model. The goal is to learn the network parameters as well as the centroids that minimize this objective. Specifically, we would like to learn a small set of centroids and encourage the network parameters to populate a small region about these centroids only.

$$\min_{W,M} \Psi(W, M) \tag{6}$$

$$\Psi(W, M) = \sum_{i=1}^{N} \|w_i - m_i\|^2 \tag{7}$$

We note that the centroids $m_i$ are correlated to $w_i$ by equation 2 with the difference that the number of centroids $k$ is also obtained through the compression process.

In the absence of any additional constraints, this optimization problem reverts back to one solved by Han et al. (2015) for Deep Compression. The solution to this problem is simply $w_i = m_i$. However, this might result in significant changes in the accuracy. We address this issue by introducing a constraint on the loss function.

$$\mathcal{L}(W) \leq \overline{\ell} \tag{8}$$

Here, $\overline{\ell}$ is the hard constraint derived based on the optimal value of the loss function. We will discuss how we determine this hyper-parameter later in this section. While solving our optimization problem, we will make sure that this condition is always satisfied.

Next, we present our method for solving the problem of minimizing the complexity of a trained model. This method iteratively solves the equation 6 for $W$ and then calculates $M$ based on equation 2, as presented in algorithm 1.

---

**Algorithm 1** Compression

---

$t \leftarrow 0$
Initialize $\bar{\ell}$
Randomly initialize $k$ centroids
**while** Convergence is not achieved **do**
    Calculate $M_t$ using equation 3.
    Solve equation 6 for $W_t$.
    Update $C$ using equation 2.
    Eliminate unnecessary centroids.
    $t \leftarrow t + 1$
**end while**

---

This algorithm first initializes $k$ random centroids. The value of $k$ is provided to the algorithm as a hyper-parameter and can be large. During the compression this algorithm eliminates or merges unnecessary centroids and reduces $k$. At the end of each iteration, if a centroid has not been assigned to any network parameter or is close to another centroids, it is eliminated.

Each iteration $t$ of the compression, finds the nearest centroid to each network parameter and solves the constrained optimization problem by finding a displacement for the model parameters like $\Delta W_t$ which solves:

$$\min_{\Delta W} \Psi(W_t + \Delta W, M_t) \tag{9}$$

$$\mathcal{L}(W_t + \Delta W) - \bar{\ell} \leq 0 \tag{10}$$

This way, the model update for the next iteration would be: $W_{t+1} = W_t + \Delta W_t$.

The solution of this constrained optimization problem should satisfy the KKT condition.

$$\nabla_{\Delta W} \Psi(W_t + \Delta W^*, M_t) + \mu \nabla_{\Delta W} \mathcal{L}(W_t + \Delta W^*) = 0 \tag{11}$$

$$\mathcal{L}(W_t + \Delta W^*) - \bar{\ell} \leq 0 \tag{12}$$

$$\mu \geq 0 \tag{13}$$

This system has two solutions. When $\mu = 0$, the solution is $\forall i : w_i^* = m_i$. If $\Delta W^*$ satisfies the constraint in equation 8, this solution is valid. Otherwise:

$$\nabla_{\Delta W} \Psi(W_t + \Delta W^*, M_t) + \mu \nabla_{\Delta W} \mathcal{L}(W_t + \Delta W^*) = 0 \tag{14}$$

$$\Rightarrow 2(W_t - M_t) + \mu \nabla_{\Delta W} \mathcal{L}((W_t + \Delta W^*)) = 0 \tag{15}$$

$$\mathcal{L}(W_t + \Delta W^*) - \bar{\ell} = 0 \tag{16}$$

$$\mu > 0 \tag{17}$$

The difficulty of solving this system is the unknown value and gradient of the loss function at the solution which we want to obtain. We get around this issue by locally estimating the loss function using its linear Taylor expansion and replacing that in the system.

$$\tilde{\mathcal{L}}(W + \Delta W) = \mathcal{L}(W) + \nabla_W \mathcal{L}(W)^T \Delta W \tag{18}$$

We then approximately calculate the displacement of the weights ($\Delta W$) with regards to $W_0$, representing the trained parameters, as below:

$$\begin{cases} 2(W_0 + \Delta W - M) + \mu \nabla_W \tilde{\mathcal{L}}(\Delta W) = 0 \\ \tilde{\mathcal{L}}(\Delta W) = \mathcal{L}(W_0) + \nabla_W \mathcal{L}(W_0)^T \Delta W \\ \tilde{\mathcal{L}}(W_0 + \Delta W) - \bar{\ell} = 0 \\ \mu > 0 \end{cases} \tag{19}$$

This system has a closed form solution:

$$\Delta W = -\frac{\mu}{2} \nabla_W \mathcal{L}(W_0) + M - W \tag{20}$$

$$\mu = 2 \frac{\mathcal{L}(W_0) + \nabla_W \mathcal{L}^T (M - W_0) - \bar{\ell}}{\|\nabla_W \mathcal{L}(W_0)\|^2} \tag{21}$$

This solution is of course valid only in a neighborhood around $W_0$ where $\Delta \tilde{W}$ is an accurate estimation of $\mathcal{L}(W_0 + \Delta W)$. We denote this region by the radius $\rho$. That is, when $\|\Delta W\| \leq \rho$. This technique is similar to local modeling of objective functions as done in Trust Region optimization methods. We also adopt a similar method as trust region techniques to initialize and update the value of $\rho$ in each iteration of the algorithm. Thus, we summarize the algorithm to solve this problem in algorithm 2.

---

**Algorithm 2** Calculate the displacement of $W$ for iteration $t$

---

$n \leftarrow 0$
Initialize $\rho_t^0$
**while** True **do**
    Solve the constrained optimization problem using equation 20 and find $\Delta W^n$
    $\lambda \leftarrow \min(1, \frac{\rho_t^n}{\|\Delta W^n\|}$
    **if** $\bar{\ell} < \mathcal{L}(W_t + \lambda * \Delta W^n)$ **then**
        $\rho_t^{n+1} \leftarrow \frac{1}{2}\rho_t^n$
    **else**
        $W_{t+1} \leftarrow W_t + \lambda * \Delta W^n$
        $\rho_{t+1}^0 \leftarrow 2 * \rho_t^n$
        break
    **end if**
**end while**

---

In this algorithm, each iteration receives a trust region radius as input and finds the displacement for $W_t$ based on equation 20. Then, if this displacement is larger than the trust region radius, the algorithm scales it. Finally, it checks if the loss function corresponding to this displacement satisfies the loss constraint. If not, it will shrink the trust region radius and retries the condition. Otherwise, it will update the parameters and expands the trust region radius. This new radius will be provided to the next iteration of Algorithm 1.

Lastly, we discuss how we obtain the value of $\bar{\ell}$. We initialize this value to the loss value corresponding to the original model $\bar{\ell} = \mathcal{L}(W_0)$. After the compression is done, we can test the accuracy of the compressed model. If this accuracy is sufficient, we can increase it. In our technique, each time we achieved a sufficient accuracy for the compressed model, we increase $\bar{\ell}$ by 20%.

## 4 MODELS

We test our compression technique on three datasets: MNIST (LeCun et al., 1998b), CIFAR-10 (Krizhevsky & Hinton, 2009), and SVHN (Netzer et al., 2011). MNIST is a collection of 6000 $28 \times 28$ black-and-white images of handwritten digits. CIFAR-10 is a set of 60000 images of object from 10 different classes. Finally, SVHN contains 60000 images of digits captured by google street view. We train a LeNet-5 on the MNIST dataset and a smaller version of the VGG network (Hubara et al., 2016) for both CIFAR-10 and SVHN. The model size and accuracy of these baseline networks are presented in table 1.

Table 1: Baseline models

| Dataset | Model | Model Size | Error Rate |
|---------|-------|------------|------------|
| MNIST | LeNet-5 (LeCun et al., 1998a) | $12Mb$ | 0.65% |
| CIFAR-10 | (Hubara et al., 2016) | $612Mb$ | 13.07% |
| SVHN | (Hubara et al., 2016) | $110Mb$ | 2.73% |

## 5 EXPERIMENTS

In this section, we experimentally study the convergence of the proposed algorithm and show that it achieves state-of-the-art compression rates. We confirm that the parameters of the compressed model cluster about the learnt centroids and evaluate the effect of this clustering on the accuracy. We then present a more in-depth study of the trade-off between accuracy and the compressed model size on our benchmarks. Finally, we compare the optimal model obtained through the trade-off analysis with the previous techniques.

The proposed compression algorithm encourages parameters to gravitate towards a set of centroids, which it learns simultaneously. We verify this on MNIST by initializing 256 random centroids. These centroids are eliminated or optimized according to Algorithm 6. Figure 1a illustrates the optimization of the centroids. In each iteration $t$, this figure shows the accuracy gap between the current model, $W_t$, and its compression using the current centroids, $C_t$. We observe that this gap decreases as the centroids are adjusted. In addition, we observe little uncompressed model accuracy degradation, despite the parameter updates. We confirm these updates cause meaningful changes to the parameters in Figure 1b. This figure depicts the distribution of the parameters before and after the compression. We can see that the parameters approach $4$ centroids, with $86\%$ of them gathered around $0$. Overall, the compression results in $112\times$ reduction in the model size with only $0.53\%$ drop in accuracy. We note that the number of pruned parameters in this result are lower compared to previous works such as Deep Compression. However, the model size reduction is higher due to the smaller dictionary.

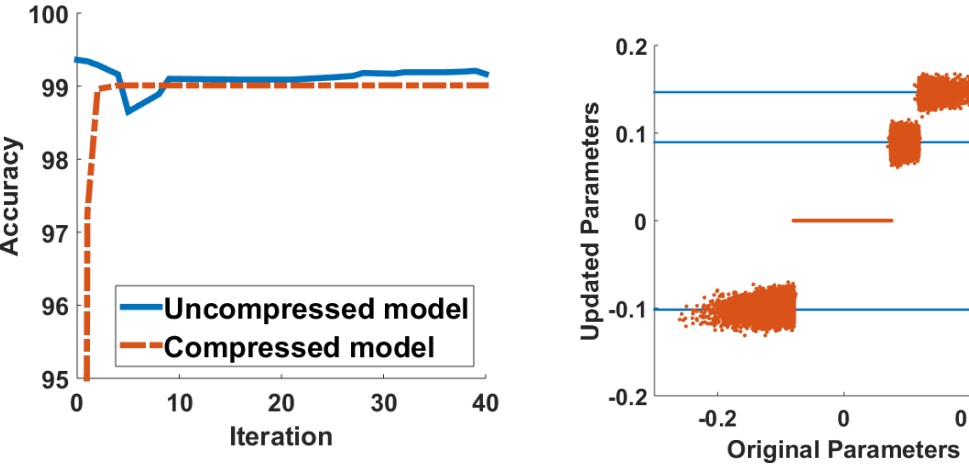

(a) Effect of reducing the complexity and learning the centroids on the uncompressed and compressed model accuracies.

(b) Final distribution of network parameters.

Figure 1: Compression of LeNet-5 using the proposed method

The loss bound (section 2) controls the final model size, introducing a trade-off between accuracy and compression. We study this trade-off by incrementing the loss bound and optimizing the model size using Algorithm 1. In the analysis for each of the benchmark datasets we initialize 256 random centroids and depict the results in Figure 2. We can see that for small reductions in size, the accuracy remains generally the same. During this stage, most clusters are quickly eliminated due to being empty. As the model size becomes small however, the error starts to increase quickly. We also observe clusters being eliminated less often. These eliminations are a result of merging clusters. Using these analyses, we can find the optimal trade-off between the accuracy and model size.

Table 2 summarizes the optimal trade-offs obtained from the previous analysis and compares them with previous works. With little drop in accuracy, the proposed method achieves state-of-the art accuracy for the MNIST dataset. It also achieves similar reductions in model size for the SVHN and CIFAR-10 datasets using the much larger network model.

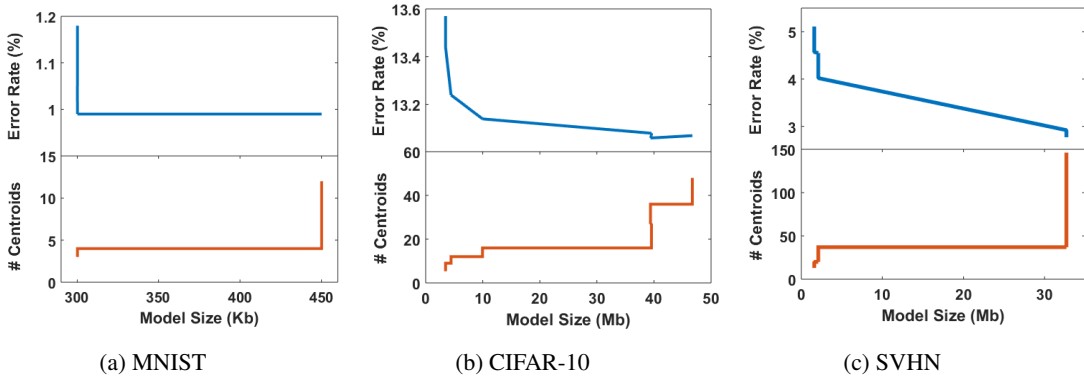

(a) MNIST            (b) CIFAR-10            (c) SVHN

Figure 2: The trade-off between accuracy and model compression for MNIST, SVHN, and CIFAR-10, using the constrained compression method.

Table 2: Baseline models

| Dataset | Compression Technique | Compression Ratio | Accuracy Loss |
|---------|----------------------|-------------------|---------------|
| MNIST | Constrained Compression | 112 | 0.53 |
| | Deep Compression | 39 | -0.06 |
| | Iterative ECSQ | 49 | -0.02 |
| | Soft Weight-Sharing | 64 | -0.05 |
| CIFAR-10 | Constrained Compression | 42 | 0.78 |
| SVHN | Constrained Compression | 128 | 2.37 |

## 6   CONCLUSION

In this paper, we presented a method for compressing trained neural network models that directly minimizes its complexity while maintaining its accuracy. For simplicity of calculations, we chose to represent the complexity using the k-means objective which is frequently used in previous works. In doing so, we maintain the accuracy by introducing a hard constraint on the loss function. This constraint incorporates the information learnt during the training process into the optimization. We then present an algorithm that iteratively finds the optimal complexity. We test this solution on several datasets and show that it can provide state-of-the-art compression, with little accuracy loss.

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
