# OpenReview forum: "DNN Model Compression Under Accuracy Constraints"
_ICLR.cc/2018/Conference — Reject_

### Official Review · AnonReviewer2 · 2017-11-25
**Promising Idea, Confusing Writing, Key Experiment Missing**

**Rating:** 4
**Confidence:** 3

**Review:**

1. Summary

This paper introduced a method to learn a compressed version of a neural network such that the loss of the compressed network doesn't dramatically change.


2. High level paper

- I believe the writing is a bit sloppy. For instance equation 3 takes the minimum over all m in C but C is defined to be a set of c_1, ..., c_k, and other examples (see section 4 below). This is unfortunate because I believe this method, which takes as input a large complex network and compresses it so the loss in accuracy is small, would be really appealing to companies who are resource constrained but want to use neural network models.


3. High level technical

- I'm confused at the first and second lines of equation (19). In the first line, shouldn't the first term not contain \Delta W ? In the second line, shouldn't the first term be \tilde{\mathcal{L}}(W_0 + \Delta W) ?
- For CIFAR-10 and SVHN you're using Binarized Neural Networks and the two nice things about this method are (a) that the memory usage of the network is very small, and (b) network operations can be specialized to be fast on binary data. My worry is if you're compressing these networks with your method are the weights not treated as binary anymore? Now I know in Binarized Neural Networks they keep a copy of real-valued weights so if you're just compressing these then maybe all is alright. But if you're compressing the weights _after_ binarization then this would be very inefficient because the weights won't likely be binary anymore and (a) and (b) above no longer apply.
- Your compression ratio is much higher for MNIST but your accuracy loss is somewhat dramatic, especially for MNIST (an increase of 0.53 in error nearly doubles your error and makes the network worse than many other competing methods: http://rodrigob.github.io/are_we_there_yet/build/classification_datasets_results.html#4d4e495354). What is your compression ratio for 0 accuracy loss? I think this is a key experiment that should be run as this result would be much easier to compare with the other methods.
- Previous compression work uses a lot of tricks to compress convolutional weights. Does your method work for convolutional layers?
- The first paper to propose weight sharing was not Han et al., 2015, it was actually:
Chen W., Wilson, J. T., Tyree, S., Weinberger K. Q., Chen, Y. "Compressing Neural Networks with the Hashing Trick" ICML 2015
Although they did not learn the weight sharing function, but use random hash functions.


4. Low level technical

- The end of Section 2 has an extra 'p' character
- Section 3.1: "Here, X and y define a set of samples and ideal output distributions we use for training" this sentence is a bit confusing. Here y isn't a distribution, but also samples drawn from some distribution. Actually I don't think it makes sense to talk about distributions at all in Section 3.
- Section 3.1: "W is the learnt model...\hat{W} is the final, trained model" This is unclear: W and \hat{W} seem to describe the same thing. I would just remove "is the learnt model and"


5. Review summary

While the trust-region-like optimization of the method is nice and I believe this method could be useful for practitioners, I found the paper somewhat confusing to read. This combined with some key experimental questions I have make me think this paper still needs work before being accepted to ICLR.

---

### Official Review · AnonReviewer3 · 2017-11-26
**The paper is clearly unqualified for publication in the current stage.**

**Rating:** 3
**Confidence:** 3

**Review:**

The paper addresses an interesting problem of DNN model compression. The main idea is to combine the approaches in (Han et al., 2015) and (Ullrich et al., 2017) to get a loss value constrained k-means encoding method for network compression. An iterative algorithm is developed for model optimization. Experimental results on MNIST, CIFAR-10 and SVHN are reported to show the compression performance.

The reviewer would expect papers submitted for review to be of publishable quality. However, this manuscript is not polished enough for publication: it has too many language errors and imprecisions which make the paper hard to follow. In particular, there is no clear definition of problem formulation, and the algorithms are poorly presented and elaborated in the context.

Pros:

- The network compression problem is of general interest to ICLR audience.

Cons:

- The proposed approach follows largely the existing work and thus its technical novelty is weak.

- Paper presentation quality is clearly below the standard.

- Empirical results do not clearly show the advantage of the proposed method over state-of-the-arts.

---

### Official Review · AnonReviewer1 · 2017-11-26
**The manuscript presents a compression method for DNN, but I cannot find significant differences from and advantages over existing deep compression approaches. Besides, the experiments are not persuasive.**

**Rating:** 3
**Confidence:** 5

**Review:**

1. This paper proposes a deep neural network compression method by maintaining the accuracy of deep models using a hyper-parameter. However, all compression methods such as pruning and quantization also have this concern. For example, the basic assumption of pruning is to discard subtle parameters has little impact on feature maps thus the accuracy of the original network can be preserved. Therefore, the novelty of the proposed method is somewhat weak.

2. There are a lot of new algorithms on compressing deep neural networks such as [r1][r2][r3]. However, the paper only did a very simple investigation on related works.
[r1] CNNpack: packing convolutional neural networks in the frequency domain.
[r2] LCNN: Lookup-based Convolutional Neural Network.
[r3] Xnor-net: Imagenet classification using binary convolutional neural networks.

3. Experiments in the paper were only conducted on several small datasets such as MNIST and CIFAR-10. It is necessary to employ the proposed method on benchmark datasets to verify its effectiveness, e.g., ImageNet.

---

### Decision · Program_Chairs · 2018-01-29
**ICLR 2018 Conference Acceptance Decision**

**Decision:**

Reject

**Comment:**

Proposed network compression method offers limited technical novelty over existing approaches, and empirical evaluations do not clearly demonstrate an advantage over current state-of-the-art.
Paper presentation quality also needs to be improved.